# *Bifidobacterium animalis* A12 and *Lactobacillus salivarius* M18-6 Alleviate Alcohol Injury by keap1-Nrf2 Pathway and Thioredoxin System

**DOI:** 10.3390/foods12030439

**Published:** 2023-01-17

**Authors:** Yan Zhang, Jingsheng Ma, Nanqing Jing, Hongxing Zhang, Yuanhong Xie, Hui Liu, Xiangfen Shan, Jianhua Ren, Junhua Jin

**Affiliations:** 1Key Food Science and Engineering College, Beijing University of Agriculture, Beijing Laboratory of Food Quality and Safety, Beijing Key Laboratory of Detection and Control of Spoilage Organisms and Pesticide Residues in Agricultural Products, Beijing 102206, China; 2Key Ningxia Saishang Dairy Co., Ltd., Yinchuan 750299, China; 3Key College of Bioengineering, Beijing Polytechnic, Beijing 100176, China

**Keywords:** alcohol injury, *Bifidobacterium animalis*, *Lactobacillus salivarius*, oxidative stress, gene expression

## Abstract

Excessive drinking can significantly damage people’s health and well-being. Although some lactic acid bacterial strains have been previously shown to alleviate the symptoms of alcohol injury, the mechanism underlying these effects remains unclear. The aim of this study was to establish an alcohol injury model and examine the protective effect and mechanism of *B. animalis* A12 and *L. salivarius* M18-6. The results showed that A12 freeze-dried powder could maintain the survival rate of mice with alcohol injury at 100%. Compared with Alco group, *L. salivarius* M18-6 dead cell improved the survival rate of mice, attenuated liver steatosis, and significantly down-regulated serum Alanine transaminase (ALT) level; at the same time, it activated keap1-Nrf2 signaling pathway and up-regulated Superoxide dismutase (SOD), it protects mouse liver cells from oxidative stress induced by alcohol injury. In addition, *B. animalis* A12 can reduce the stress response to short-term alcohol intake and improve the ability of anti-oxidative stress by upregulating the level of isobutyric acid, reducing the level of keap1 protein in the liver of mice and upregulating the expression of thioredoxin genes (Txnrd1, Txnrd3, Txn1). Taken together, the results showed that *B. animalis* A12 and *L. salivarius* M18-6 alleviate alcohol injury in mice through keap1-Nrf2 signaling pathway and thioredoxin system.

## 1. Introduction

Excessive dosage and duration of alcohol consumption can cause alcohol injuries. Alcohol injury has a wide range of effects, affecting the stomach, intestines, brain, liver, nervous system, reproductive system, immune system, and other bodily functions and systems [1,2,3,4]. As the problem of alcohol injury becomes increasingly serious, there is an increasing amount of research in academia and it has gradually become a hot topic. However, people are still helpless because of insufficient understanding of some alcohol injury problems, such as alcoholic liver disease (ALD), for which there is still no Food and Drug Administration (FDA)-approved treatment at any stage [5]. Alcohol metabolism in the human body involves oxidation of ethanol to acetaldehyde and further oxidation to Nicotinamide adenine dinucleotide (NADH) and reactive oxygen species (ROS) [6]. Ethanol is oxidised to acetaldehyde, mainly in the cytoplasm of the liver, by ethanol dehydrogenase and its coenzyme nicotinamide adenine dinucleotide [7].

Lactic acid bacteria (LAB) is a general term for non-spore, non-pathogenic, and gram-positive bacteria that can convert carbohydrates (such as glucose, fructose and lactose) into lactic acid. *Bifidobacterium* and *Lactobacillus* are common probiotics that are used in functional foods. In recent years it has been found that some probiotics have a significant alleviating effect on ALD [8]. With the in-depth research of probiotics on alcohol injury, people have found that alcohol injury outside of ALD also has a very good performance [9,10]. The mechanism of action of probiotics on alcohol injury found so far mainly includes reducing oxidative stress [11], lowering endotoxin levels, regulating certain cytokine metabolism levels [12], improving the environment of intestinal flora, and repairing intestinal barrier function and the mucosal immune system, etc. [13,14,15]. Studies have shown that probiotics can reduce acute alcoholic liver injury in mice by activating alcohol metabolism enzymes and reducing liver TNF-α response [16]. For example, *L. plantarum* attenuates alcohol-induced epithelial tight junction and endotoxemia through EGF receptor-dependent mechanisms [17]. *L. paracasei* GKS6, *L. plantarum* GKM3, and *L. rhamnosus* GKLC1 can reduce alcoholic liver disease in mice [18], while *L. fermentum* LA12 can significantly reduce the production and permeability of nitric oxide in the intestinal tract of rats with alcohol-induced injury, up-regulate the mRNA expression level of tight junction protein, thereby stimulating the recovery of barrier structure and function, preventing endotoxin from infiltrating into the blood [19]. Additionally, *L. rhamnosus* CCFM1107 alleviates alcohol injury by reducing oxidative stress, and *L. rhamnosus* granules can balance intestinal microbiome disorder and improve chronic alcohol-induced liver injury [20]. The application of *B. animalis* and *L. salivarius* to alcohol injury in keap1-Nrf2 signaling pathways and thioredoxin systems has not been reported. Therefore, this study has important significance.

The National Institute on Alcohol Abuse and Alcoholism (NIAAA) model is a new alcohol injury model, which can highly simulate the pathogenesis of human alcoholic hepatitis. This results in a significant increase in the level of serum ALT, obvious steatosis and inflammation in the liver, and high concentration of alcohol in serum [21,22,23]; therefore, it has a good modelling effect. There are few reports on probiotics in this model. *B. animalis* A12 isolated from the faeces of breast-fed healthy infants has potential hypoglycaemic and lipid-lowering effects [24]. Previous studies have proved that the *Bifidobacterium* has a series of excellent probiotic characteristics. M18-6 is an *L. salivarius* presented good antioxidant activity in our previous study [25]. Therefore, this study selected *B. animalis* A12 and *L. salivarius* M18-6 and explored the mechanism of action of the two probiotics on alcohol injury from the perspectives of oxidative stress and immunity.

## 2. Materials and Methods

*B. animalis A12* (CGMCC No.17308), *L. salivarius* M18-6 (separation and preservation of bacterial strains in the functional dairy laboratory of Beijing University of Agriculture, CGMCC No.16199), and *B. animalis* A12 freeze-dried powder were used in this study.

*Bifidobacterium* culture medium was created as follows: add 0.5 g/L of L-cysteine salt (Beijing Chemical Plant) on the basis of Man Rogosa Sharpe (MRS) medium, pH adjusted to 6.5, aliquot into an anaerobic tube, seal it with nitrogen and autoclave at 121 °C for 15 min.

### 2.1. Microbiology Experiment

*B. animalis* A12 and *L. salivarius* M18-6 were taken from an −80 °C bacteria bank and inoculated in MRS liquid medium for amplification and culture to the third generation. After counting, the cell suspension concentration was diluted to 5.0 × 10^9^ CFU/mL with normal saline. To get the dead cells used in this study, the living bacteria were sterilised at 121 °C for 15 min, centrifuged at 4000× *g* and 4 °C for 15 min, and the concentration of cell suspension was adjusted using normal saline.

### 2.2. Animal Experimental Design

The animal experiments were performed in accordance with the recommendations of Animal Management Regulations, Ministry of Science and Technology of the People’s Republic of China. The protocol was approved by the Ethical Committee of the Experimental Animal Care of Beijing University of Agricultural (protocol code bua2022002, 2 March 2022, Beijing, China). Studies were conducted on 8- to 9-week-old male, C57BL/6N mice raised in the clean laboratory animal room of the Pony Testing Group Co., Ltd. The animals were housed in standard cages and maintained under standard laboratory conditions, at ambient temperatures of 20–23 °C, under a 12 h/12 h light/dark cycle, with a relative humidity of 50–60%, and with free access to water and food, Lieber-Decarli liquid feed feeding. All the mice were allowed to adapt to the set environment for 7 days and then randomised into seven groups (n = 12 per group): Normal group (fed with normal saline); Alco group (fed with alcohol liquid feed only); A12-L group (fed with live *B. animalis* A12 cells); A12-D group (fed with dead *B. animalis* A12 cells); A12-P group (fed with *B. animalis* A12 freeze-dried power); M18-6-L group (fed with live *L. salivarius* M18-6 cells); and M18-6-D group (fed with dead *L. salivarius* M18-6 cells). The formal experiment was divided into three stages: stage 1: The mice were given a liquid diet adaptation period for 1 d with no alcohol in the diet; stage 2: The alcohol concentration in the liquid diet increased from 1 to 4% in the first 4 days; and stage 3: The mice were fed with 5% alcohol liquid feed for 10 days, and then, on the morning of the 16th day, a high-dose alcohol was administered by gavage. After 9 h, the animals were processed, and blood and liver samples were taken. The body weights of the mice were measured weekly using a digital scale. Each mouse was gavaged with normal saline or the corresponding bacterial test substance once daily. The gavage volume was 0.2 mL with 1 × 10^9^ CFU live cells or 1 × 10^9^ CFU dead cells (Figure 1). 

### 2.3. Serum Biochemistry Determination

Blood was coagulated at 20 °C for one hour and then centrifuged at 3500× *g* and 4 °C for 15 min. The centrifuged serum was used immediately or stored in a low temperature environment of −70 °C for later use. An automatic biochemical analyser type 7600 (Hitachi High-tech International Trading, Shanghai, China) was used to determine mouse serum alanine transaminase (ALT).

### 2.4. Liver Biochemistry Determination

Liver homogenate (10%) was centrifuged at 3000× *g* and 4 °C for 15 min, the precipitate was discarded, the supernatant was collected, and the liver homogenate was stored at −70 °C for the experiment. Total superoxide dismutase (T-SOD) levels were determined (Nanjing Jiancheng Biotechnology Co., Ltd, Nanjing, China; Project No.: A001-1).

### 2.5. Gene Expression Analysis

Txnrd1, Txnrd3, Txn1, Txn2, Nrf2, keap1, and p65 transcript expression levels in the liver tissue were detected by real-time fluorescent quantitative PCR (Applied Biosystems, Foster City, CA, USA). RNA was extracted from liver tissue according to the manufacturer’s instructions, and the TAKARA reverse transcription kit-RR047A was used to obtain complementary DNA (cDNA). The reverse-transcribed cDNA was used as a template, β-actin acted as an internal reference gene, and specific primers (Sangon Biotech, Shanghai, China) of the target genes were synthesised according to the sequences in Table 1. A fluorescence quantitative PCR amplification system (cDNA template: 1 μL, Primer F: 0.5 μL, Primer R: 0.5 μL, TB Green: 7.6 μL, Rox: 0.4 μL (Takara Biomedical Technology Co., Ltd, Beijing, China), ddH2O: 10 μL). The PCR conditions were as follows: 5 min at 95 °C, 40 cycles of 10 s at 95 °C, 30 s at 60 °C, and 30 s at 72 °C followed by a melting curve step. The mRNA expression level was calculated using the 2^−△△CT^ method and the data were analysed.

### 2.6. Western Blot

Total protein of the tissue samples was extracted according to the manufacturer’s instructions and electrophoresis was performed on a 12% gel prepared according to the size of the target protein. The sample was loaded, and electrophoresis was conducted at 80 V for approximately 20 min. After the protein entered the separation gel and the marker began to separate, the voltage was adjusted to 120 V. The total electrophoresis time was approximately 1.5 h. The membrane was then transferred using a wet transfer method. The transferred membrane was then shaken slowly in a blocking solution at room temperature for 1 h. After blocking, the membrane was washed with TBST (10X TBST (100 mL/L): Tris (24.2 g), and NaCl (80 g). Water was added to adjust the pH to7.6 and dilute the volume to 1 L (Tween-20 1 mL/L), incubated with the primary antibody (keap1 dilution ratio of 1:1000, Nrf2 dilution ratio of 1:500, Proteintech, Chicago, USA, β-actin dilution ratio of 1:5000, Powan Biotechnology, Shanghai, China) in a refrigerator at 4 °C, and then incubated with the secondary antibody at room temperature. After the incubation, the membrane was washed with TBST, to allow the colour to develop, and photos were taken of the exposure at different times to detect the protein expression of Nrf2 and keap1 in liver tissues.

### 2.7. Haematoxylin and Eosin (H&E) Staining of Mouse Liver

Liver tissues were analysed using Beijing Xuebang Technology Co., Ltd. The liver lobules of each group were fixed in a solution of 4% paraformaldehyde solution, dehydrated in alcohol, and embedded in paraffin. Four-micrometre sections were prepared and stained with H&E for histological evaluation.

### 2.8. GC-MS Detection of Short-Chain Fatty Acids in Mouse Faeces

A 50 ± 1 mg sample was placed into a 2 mL Eppendorf tube, extracted with 0.5 mL dH_2_O, by vortex mixing for 10 s. The sample was then homogenized in a ball mill for 4 min at 40 Hz and ultrasound treated for 5 min (incubated in ice water). It was then centrifuged for 20 min at 5000 rpm and 4 °C. The supernatant (0.3 mL) was transferred into fresh 2 mL EP tubes, extracted with 0.5 mL dH_2_O, vortexed for 10 s, homogenised for 4 min at 40 Hz, and ultrasonicated for 5 min (incubated in ice water). It was then centrifuged for 20 min at 5000 rpm and 4 °C. After this, the supernatant (0.5 mL) was transferred into a fresh 2 mL EP tube, and the supernatant was combined with the following for a total volume of 0.8 mL; 0.1 mL 50% H_2_SO_4_ and 0.8 mL of 2-methylvaleric acid (25 mg/L stock in methyl tert-butyl ether, added as an internal standard). The sample was then vortex mixed for 10 s, oscillated for 10 min, then ultrasound treated for 5 min (incubated in ice water). It was then centrifuged for 15 min at 10,000 rpm (4 °C) and kept at −20 °C for 30 min. Finally, the supernatant was transferred to a fresh 2 mL glass vial for GC-MS analysis with a Shimadzu GC2030-QP2020 NX gas chromatography-mass spectrometer. The system used an HP-FFAP capillary column. A 1μL aliquot of the analyte was injected in split mode (5:1). Helium was used as the carrier gas, the front inlet purge flow rate was 3 mL min^−1^, and the gas flow rate through the column was 1 mL min^−1^. The initial temperature was maintained at 80 °C for 1 min, then raised to 200 °C at a rate of 10 °C min^−1^ for 5 min, then kept for 1min at 240 °C at a rate of 40 °C min^−1^. The injection, transfer line, quad, and ion source temperatures were 240, 200, and 150 °C, respectively. The energy was −70 eV in the electron impact mode. Mass spectrometry data were acquired in Scan/SIM mode with an m/z range of 33–150 after a solvent delay of 3.5 min.

### 2.9. Statistical Analysis

All data were analysed using the statistical software IBM SPSS Statistics 26 (IBM, Chicago, USA) for homogeneity test of variance, one-way analysis of variance, Duncan’s method for multiple comparisons, nonparametric test, and the t-test for significance analysis. The significance level was set at *p* < 0.05. The results of each index are expressed as “mean ± standard deviation” (x ± SD). The graphics were performed using GraphPad Prism (GraphPad Software 8, Santiago, USA). All experiments were repeated three times, except for animal experiments. For biochemical parameters, n = 6 per group was employed, whereas n = 3 per group was used for Western blot analysis.

## 3. Results

### 3.1. Effect of B. Animal A12 and L. salivarius M18-6 on Morphology and Survival Rate of Mice with Alcohol Injury

We found that after treatment with alcohol, mice showed symptoms such as body stiffness, slow movement, mental fatigue, or intense combat, and their fur began to lose its lustre. Compared with the normal group, the mice in the other groups had poor appetite and weight loss, but these symptoms were significantly improved in the other groups compared with the Alco group.

To study the protective effect of *B. animal* A12 and *L. salivarius* M18-6 on alcohol injury, we established a new alcohol injury model of NIAAA. Changes in the survival rate of the mice during the experiment are shown in Figure 2. The survival rate on the eleventh day in the Alco, M18-6-L, M18-6-D, and A12-D groups was lower than that of the normal group, but the survival rate of the A12-P group remained 100%. The survival rate was significantly higher in A12-P group than Alco group (*p* < 0.05, Figure 2D). This suggests that *B. animalis* A12 and *L. salivarius* M18-6 are beneficial to the survival rate of experimental mice, thereby alleviating alcohol-induced death. 

### 3.2. Effect of B. animalis A12 and L. salivarius M18-6 on Liver Histopathology in Mice with Alcohol Injury

Figure 3 shows a typical pathological section of the liver stained and liver index. Owing to the stimulation and destruction of the metabolic system by alcohol, the liver, as the main metabolic organ, has obvious pathological damage, such as fatty degeneration and vacuolar degeneration. It can be seen that the structure of hepatic lobule in each group is clear, the stem cells in the normal group are round, and only individual cells are fatty degeneration (Figure 3A). Liver histological damage and fatty degeneration were obvious in the Alco group (Figure 3B). Liver injury in the M18-6-L and M18-6-D groups was significantly improved, such as reduced steatosis, which changed from vacuolar degeneration to granular degeneration (Figure 3F,G), while only slight steatosis was observed in the A12-D group (Figure 3D). After alcohol intake, the liver is injured and the liver index often decreases, which can be used as a preliminary indicator of the degree of injury, and it can be seen from Figure 3H that compared with the normal group, the liver indices of the Alco group decreased with no significant difference (*p* > 0.05). The liver index in the M18-6-L group increased compared with that in the Alco group, and with a remission rate of 2.6%. These results show that M18-6-L were effective in preventing alcohol injury.

### 3.3. Effect of B. animalis A12 and L. salivarius M18-6 on Serum ALT in Mice with Alcohol Injury

We examined the effects of treatment interventions expression of alanine transaminase (ALT) in mice. As shown in Figure 4, compared with the normal group the ALT level in the Alco group was significantly increased (*p* < 0.05), indicating that the liver injury in these mice was severe. Both the A12-L and M18-6-D groups significantly alleviated the increase in serum ALT activity caused by alcohol injury in mice (*p* < 0.05). M18-6-D group serum ALT activity remission rate (=(Alco group-Experimental group)/(Alco group-Normal group) × 100%) was 37.59%. It has been suggested that the elevation of serum ALT activity caused by alcohol injury can be effectively alleviated by M18-6 dead cells.

### 3.4. Effect of B. animalis A12 and L. salivarius M18-6 on Immune Level and Liver Oxidative Status in Mice

Alcohol injury can lead to an abnormal immune system and reduced metabolic detoxification of alcohol, thereby inducing alcoholic liver disease, p65 is a subunit of the NF-kB pathway. As the central factor of the immune response, the relative expression of mRNA in each group is shown in Figure 5A. The relative expression of p65 mRNA in the Alco group was significantly lower than that in the normal group (*p* < 0.05), indicating that alcohol intake caused a stress response. However, after the intake of probiotics, compared with the Alco group the fold change in p65 was significantly increased (*p* < 0.05). The A12-D (1.26) and M18-6-D (0.99) groups restored the relative expression of p65 mRNA in the livers of mice in the normal (1) group. This study also measured SOD activity (Figure 5B); the SOD activity of the Alco group decreased compared with the normal group, while the A12-P, M18-6-D, and M18-6-L groups’ SOD activity significantly increased (*p* < 0.05), indicating that *B. animalis* A12 and *L. salivarius* M18-6 can improve the body’s antioxidant status.

### 3.5. Effect of B. animalis A12 and L. salivarius M18-6 on Short Chain Fatty Acid Content

Short-chain fatty acids (SCFA) could affect intestinal permeability and maintain normal physiological function of the gut. As shown in Figure 6A, there was a significant difference between the normal and Alco groups (*p* < 0.05), the level of valeric acid in the A12 dead cells, and freeze-dried power was higher than that in the Alco group. Figure 6B shows the isobutyric acid content in the mouse colon, where it can be clearly seen that there are significant differences between the Alco group and the normal group (*p* < 0.05), additionally, A12-D and A12-P groups significantly higher than in the Alco group (*p* < 0.05) (Figure 6B).

### 3.6. Effect of B. animalis A12 and L. salivarius M18-6 on Gene Expression of Thioredoxin System in Mice

qPCR analysis was used to determine the relative expression of these parameters at the mRNA level. As shown in Figure 7A, the relative expression levels of Txnrd1 mRNA in the livers of mice in the Alco, A12-L, A12 -D, M18-6-L and M18-6-D groups were significantly higher than those in the normal group (*p* < 0.05), and those in the A12-L and A12-P groups were significantly higher than those in the Alco group (*p* < 0.05). Therefore, *B. animalis* and *L. salivarius* can upregulate the expression of Txnrd1 in the liver and respond better to alcohol-induced oxidative damage. As shown in Figure 7B, the A12-L, A12-P, and M18-6-L groups alleviated the decrease in the relative expression of Txnrd3 mRNA in the livers of mice with alcohol injury. The M18-6-L group restored the relative expression of Txnrd3 mRNA in mouse liver to the normal group, and the effect was the best for this group. Compared with the Alco and normal groups, the expression of Txn1 in the A12-L group was significantly increased (*p* < 0.05; Figure 7C). This indicated that A12 living cells could improve the body environment, enhance antioxidant capacity, and resist alcohol injury by increasing the relative expression of Txn1 mRNA in the mouse liver. As shown in Figure 7D, the A12-L and A12-D groups reduced the stress response to short-term alcohol intake and reversed the relative expression of Txn2 mRNA in the liver of mice, returning to the control level without alcohol treatment. The A12-D group showed the best results.

### 3.7. Effect of B. animalis A12 and L. salivarius M18-6 on Immune Proteins in Mice with Alcohol Injury

The oxidative stress ability of the liver can be evaluated by measuring the levels of immune factor proteins in the livers of mice. When the liver is damaged the level of immune factor proteins changes, indicating that the liver may be damaged and that it requires timely intervention. Figure 8B shows that the relative expression level of the keap1 protein in the Alco group was significantly higher than that in the normal group (*p* < 0.05), indicating that alcohol intervention increased the expression of keap1 in liver tissue and with liver damage. The relative expression levels of keap1 in the A12-D, A12-P, and M18-6-D groups were significantly lower than those in the Alco group (*p* < 0.05). Therefore, treatment with A12 dead cells, A12 freeze-dried powder, or M18-6 dead cells can reduce the expression of keap1 and alleviate alcohol injury. Nrf2 has been shown to have anti-inflammatory effects. As shown in Figure 8C, compared with the normal group the expression level of the Nrf2 protein in the alcohol group was increased (*p* < 0.05). The expression of Nrf2 protein in the A12-L, A12-D, and A12-P groups was significantly decreased, indicating that A12 living cells, dead cells, and freeze-dried powder alleviated oxidative damage in mice. These results suggest that the protective effect of probiotics in alcohol-injured mice is correlated with the keap-Nrf2 signaling pathway.

## 4. Discussion

As current studies have shown, not all probiotics can prevent and treat alcohol injury, and some can even aggravate the disease. Live cells are beneficial to the host as they can survive and colonize in the animal digestive tract, compete with pathogenic bacteria, and produce useful enzymes and metabolites; after inactivation, the active components of the cell wall, such as exopolysaccharides, peptidoglycan and lipoteichoic acid, will be dissociated from the cell wall membrane and may play an active role in some functions. Freeze-dried powder uses multi-layer embedding technology to have high survival rate, protect strains from reaching the intestine alive, adhere to the intestine, and drive away harmful bacteria. Therefore, we evaluated the effects of *B. animalis* A12 live cells, dead cells, freeze-dried powder, *L. salivarius* M18-6 live cells, and dead cells on the pathology, immune system, antioxidant system, and intestinal content metabolites. Furthermore, this study aimed to explore whether these factors have positive effects on the prevention and treatment of alcohol injury, and to analyse its mechanism.

Cuicui et al. established a mouse model of acute liver injury, studied the hepatoprotective effect of *L. plantarum* C88, and determined the survival rate of mice [26]. The results showed that no symptoms of death were observed in the group treated with *L. plantarum* C88, similar to the results of this study, where the survival rate of mice in the *B. animalis* A12 and *L. salivarius* M18-6 treatment groups was significantly improved, indicating that the test substances involved in *B. animalis* A12 and *L. salivarius* M18-6 played a positive role in ensuring the survival of the mice. In particular, during the third phase of the experiment A12 freeze-dried powder maintained alcohol-induced survival in mice.

ALT and AST are considered indicators of liver disease. Hepatocyte injury often leads to the infiltration of ALT and AST into the blood, which then increases serum transaminase levels. The degree of alcoholic liver injury was evaluated by measuring serum ALT levels. H&E staining can directly reflect histopathological changes in liver tissue. A previous study showed that probiotic treatment could effectively reduce alcohol injury in mice, which was similar to the results of this study. At the same time, compared with the normal group, the serum ALT level of the Alco group was significantly increased, which was consistent with previous reports [27]. In this study, dead *L*. *salivarius* M18-6 cells most significantly reduced the ALT increase caused by alcohol injury, followed by *B. animalis* A12 live cells. In addition, H&E staining of liver sections showed that steatosis and vacuolisation of the alcohol group were the most serious, and the probiotic group was significantly improved. Previous studies have shown that LAB intervention can reduce pathological damage to liver tissue [28]. In this study, SOD levels decreased in the Alco group. However, SOD levels in *B. animalis* A12 freeze-dried powder and *L. salivarius* M18-6 live and dead cells returned to higher levels than those in the normal group. It has been reported that probiotics can increase the SOD level of the liver tissue in mice [29,30,31], which is similar to our results. This shows that the *B. animalis* A12 and *L. salivarius* M18-6 in this study play an important role.

SCFA are currently considered one of the most important metabolites of intestinal flora, mainly produced in the caecum and colon, and are an important source of energy for the host. In the chronic stages of various metabolic diseases, SCFA play an important role in reducing inflammation, promoting lipid accumulation in adipocytes, reducing inflammatory fat infiltration, and improving the intestinal mucosal barrier [32]. In the meantime, it could affect intestinal permeability and maintain normal physiological function of the gut [33,34]. Studies have reported [35] that *L. fermentum* can increase SCFA in the colon contents of colitis mice. Recent studies have also shown that butyrate helps alleviate alcohol-induced gastric ulcers, and studies have shown that SCFA have beneficial effects in alcohol-induced mice [36,37]. These results are similar to those of the present study. The results showed that *B. animalis* A12 dead cells and freeze-dried powder significantly increased the level of isobutyric acid in the intestinal contents, enhanced the viability of intestinal cells, reduced fat infiltration, and regulated immune function, thereby reducing alcohol injury.

The thioredoxin system is an important intracellular oxidation system that consists of NADPH, Trx, and TrxR. Its function is to provide an effective electron donor for ribonucleic acid reductase [38], which plays a role in inflammation, apoptosis, and cardiovascular diseases [39]. TrxR1 and TrxR2 play important roles in reducing cellular oxidative stress injury and scavenging mitochondrial cytoplasmic free radicals. Trx plays a major role in the thioredoxin system, which is considered a biomarker of oxidative stress [40]. Trx jointly scavenges ROS by interacting with peroxide oxidoreductase, thereby playing an anti-oxidative stress role [41]. Txn1 and Txn2 are involved in Trx coding. Txnrd1 can promote the body to induce an oxidative stress response by upregulating the level of p65 protein in NF-kB and is also positively regulated by Nrf2. When the thioredoxin system is activated under oxidative stress, Trx and TrxR levels are significantly elevated to combat reactive oxygen species and alleviate damage to the body. When the liver is impaired in alcohol metabolism, gene expression in the liver (such as Txnrd1, Txnrd3, Txn1, and Txn2) decreases significantly, indicating the risk of liver disease and the need for timely intervention. Our results showed that the body entered the oxidative stress state of self-protection and began to increase the relative expression of Txnrd1 mRNA in the liver of mice, while *L. salivarius* M18-6 can further increased the relative expression of Txnrd1 mRNA in the liver to better cope with alcohol-induced oxidative damage. *B. animalis* A12 live cells and *L. salivarius* M18-6 live and dead cells significantly alleviated the relative expression of Txnrd3 mRNA in the livers of mice caused by alcohol injury. *L. salivarius* M18-6 live cells restored the relative expression of Txnrd3 mRNA in the livers of mice to the level observed in the normal group. *B. animalis* A12 live and dead cells reduced the body’s stress response to short-term alcohol intake and significantly reversed the relative expression of Txn2 mRNA in the livers of mice to the level of the normal group.

Nrf2 is usually located in the cytoplasm and is degraded by keap1, which is the main transcription factor that controls antioxidant production to maintain the redox homeostasis of cells [42]. Keeping high sensitivity to ROS and RNS [43] when cells are stimulated, Nrf2 can induce the transcription of anti-oxidative stress genes and phase II detoxification enzymes [44], improving cell self-protection [45]. When a disease occurs, the interaction between keap1 and Nrf2 is interrupted by various stimuli. Activated Nrf2 enters the nucleus by binding to antioxidant response elements (AREs), stimulating several related cell protections and antioxidant gene transcriptions. The Nrf2/ARE signalling pathway is a key mediator of general redox and tissue-specific homeostasis [46]. Keap1 is an excellent redox sensor for biological and intracellular oxidative stress. Under normal pressure-free conditions, Nrf2 levels in cells are very low, and Nrf2 and keap1 form complexes in the cytoplasm. Keap1 targets Nrf2 for multi-ubiquitination through ligase 3 followed by proteasomal degradation. The Nrf2/ARE pathway plays an important role in cancer treatment [47] and enhancing the ability of cells to resist oxidative stress [48]. It has been reported that lactic acid bacteria resident in the intestinal tract can activate Nrf2 in the liver and prevent oxidative liver injury [49]. It has also been reported that *L. fermentum* can activate the Nrf2 signaling pathway and upregulate the expression of Nrf2 in the liver. The results of this experiment showed that alcohol increased the expression of the negative regulatory factor keap1. Compared with the Alco group, the protein expression of keap1 in mice of *B. animalis* A12 dead cells, freeze-dried power, and *L. salivarius* M18-6 dead cell groups was significantly decreased, while the of the *L. salivarius* live cells and dead cells groups was significantly increased. This suggests that *L. salivarius* activates Nrf2 signalling and alleviates alcohol injury, possibly by preventing oxidative liver injury. However, this is only a preliminary discussion and the specific causal relationship requires further verification. In the future, we will continue to explore the changes in intestinal flora in mice with alcohol injury, and even the clinical trials of *B. animalis* A12 and *L. salivarius* M18-6 for the treatment of alcohol injury are needed to provide definitive evidence on its therapeutic efficacy.

## 5. Conclusions

In this study, our findings showed that *B. animalis* A12 and *L. salivarius* M18-6 had preventive and protective effects against alcohol injury in mice. *L. salivarius* M18-6 dead cells protect mouse hepatocytes from alcohol-induced oxidative stress injury by down-regulating serum ALT level, activating keap1-Nrf2 signaling pathway and up-regulating SOD. In addition, *B. animalis* A12 can reduce the protein level of keap1 in the liver and up-regulate the expression of thioredoxin genes, reduce the stress response to short-term alcohol intake, and improve the ability of antioxidant stress by upregulating the level of isobutyric acid. The results of this study provide a scientific basis for the clinical application and product development of probiotics for alcohol injury.

## Figures and Tables

**Figure 1 foods-12-00439-f001:**
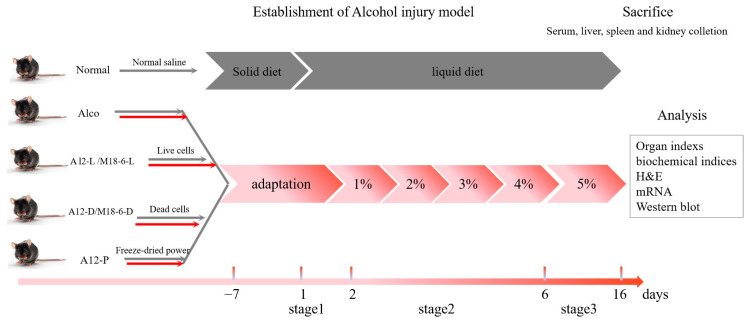
The Schedule of the experimental protocol and drug administration. (n = 12 per group). Normal: normal saline-fed; Alco: alcohol-fed group; A12-L: alcohol-fed with live *B. animalis* A12 cells group, A12-D: alcohol-fed with dead *B. animalis* A12 cells group, A12-P: alcohol-fed with *B. animalis* A12 freeze-dried power group, M18-6-L: alcohol-fed with live *L. salivarius* M18-6 cells group, and M18-6-D: alcohol-fed with dead *L. salivarius* M18-6 cells group.

**Figure 2 foods-12-00439-f002:**
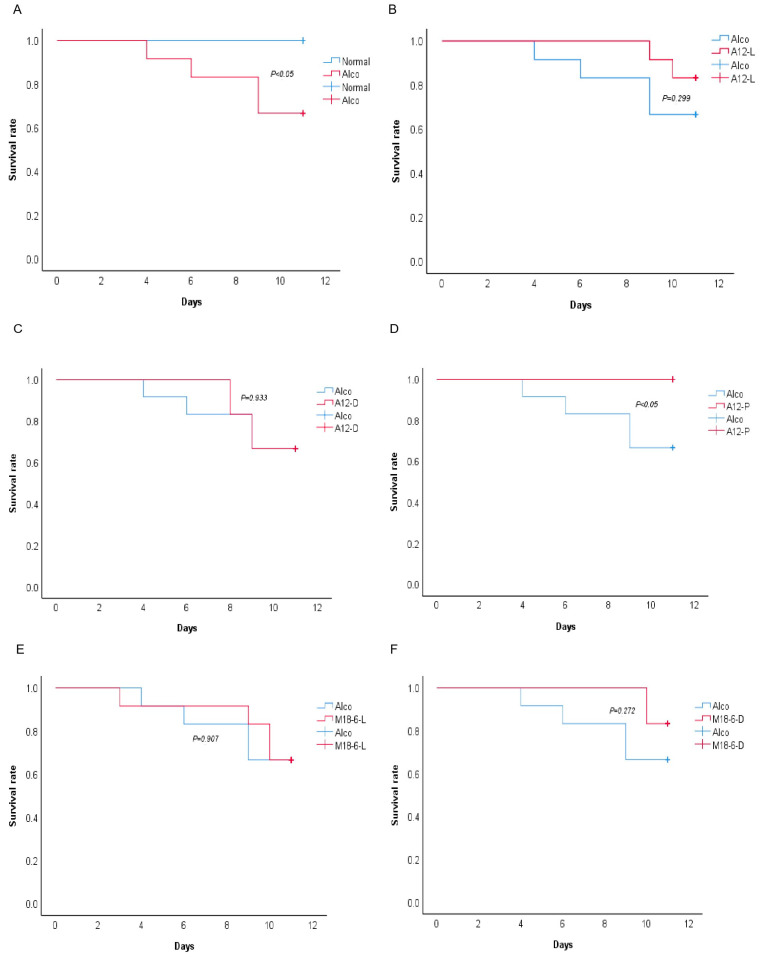
The Kaplan–Meier survival curves in different treated groups. The *p*-values were calculated using the log-rank test. Each graph shows the comparison of survival rate. (**A**): Between normal and Alco; (**B**): Between Alco and A12-L; (**C**): Between Alco and A12-D; (**D**): Between Alco and A12-P; (**E**): Between Alco and M18-6-L; (**F**): Between Alco and M18-6-D. Normal: normal saline-fed; Alco: alcohol-fed group; A12-L: alcohol-fed with live *B. animalis* A12 cells group, A12-D: alcohol-fed with dead *B. animalis* A12 cells group, A12-P: alcohol-fed with *B. animalis* A12 freeze-dried power group, M18-6-L: alcohol-fed with live *L. salivarius* M18-6 cells group, and M18-6-D: alcohol-fed with dead *L. salivarius* M18-6 cells group.

**Figure 3 foods-12-00439-f003:**
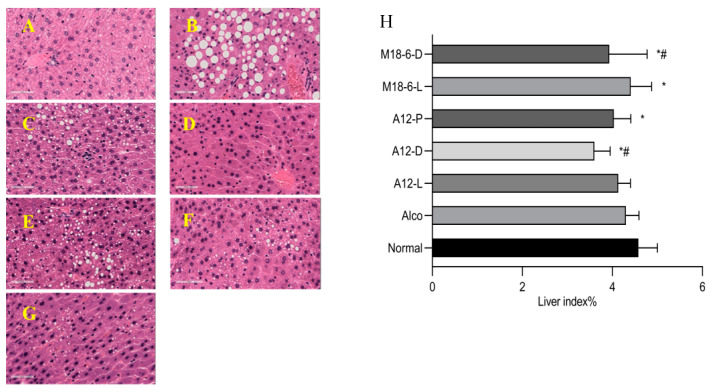
Effects of probiotics treatment on liver histopathology (H&E staining, magnification × 40) and liver index. Liver histopathology: (**A**): Normal; (**B**): Alco; (**C**): A12-L; (**D**): A12-D; (**E**): A12-P; (**F**): M18-6-L; (**G**): M18-6-D; (**H**): Liver index. Compared with normal group (*p* < 0.05) *, compared with Alco group (*p* < 0.05) #. Normal: normal saline-fed; Alco: alcohol-fed group; A12-L: alcohol-fed with live *B. animalis* A12 cells group, A12-D: alcohol-fed with dead *B. animalis* A12 cells group, A12-P: alcohol-fed with *B. animalis* A12 freeze-dried power group, M18-6-L: alcohol-fed with live *L. salivarius* M18-6 cells group, and M18-6-D: alcohol-fed with dead *L. salivarius* M18-6 cells group.

**Figure 4 foods-12-00439-f004:**
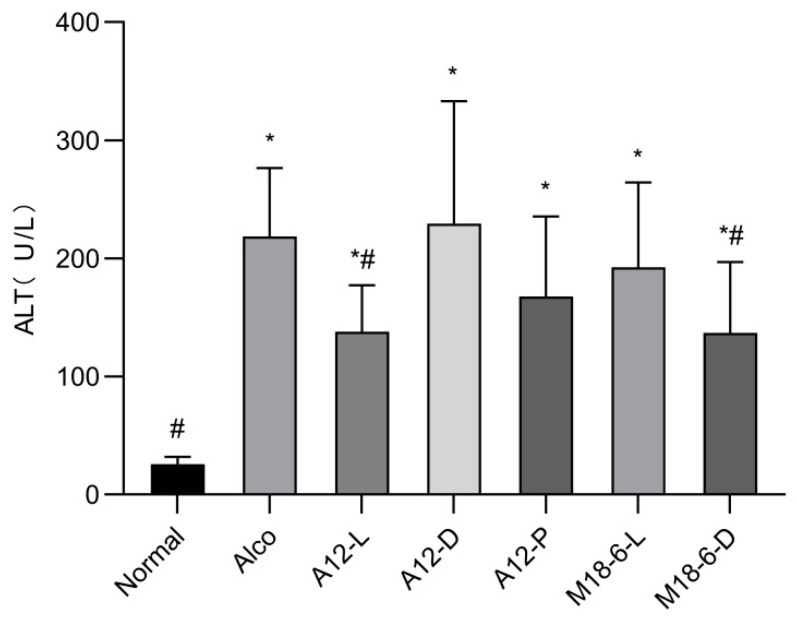
Serum ALT level in mice. Compared with normal group (*p* < 0.05) *, compared with Alco group (*p* < 0.05) #. ALT: alanine transaminase. Normal: normal saline-fed; Alco: alcohol-fed group; A12-L: alcohol-fed with live *B. animalis* A12 cells group, A12-D: alcohol-fed with dead *B. animalis* A12 cells group, A12-P: alcohol-fed with *B. animalis* A12 freeze-dried power group, M18-6-L: alcohol-fed with live *L. salivarius* M18-6 cells group, and M18-6-D: alcohol-fed with dead *L. salivarius* M18-6 cells group.

**Figure 5 foods-12-00439-f005:**
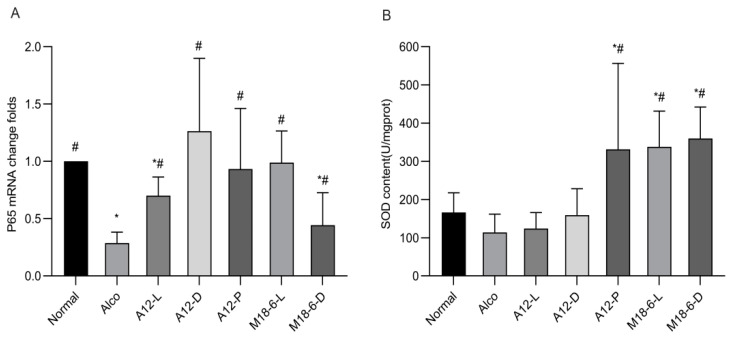
Liver p65 mRNA expression and SOD level. (**A**): Liver p65 mRNA expression level. Compared with normal group (*p <* 0.05) *, compared with Alco group (*p <* 0.05) #. (**B**): Liver SOD level in mice. p65: Transcription factor p65, SOD: Superoxide dismutase. Normal: normal saline-fed; Alco: alcohol-fed group; A12-L: alcohol-fed with live *B. animalis* A12 cells group, A12-D: alcohol-fed with dead *B. animalis* A12 cells group, A12-P: alcohol-fed with *B. animalis* A12 freeze-dried power group, M18-6-L: alcohol-fed with live *L. salivarius* M18-6 cells group, and M18-6-D: alcohol-fed with dead *L. salivarius* M18-6 cells group.

**Figure 6 foods-12-00439-f006:**
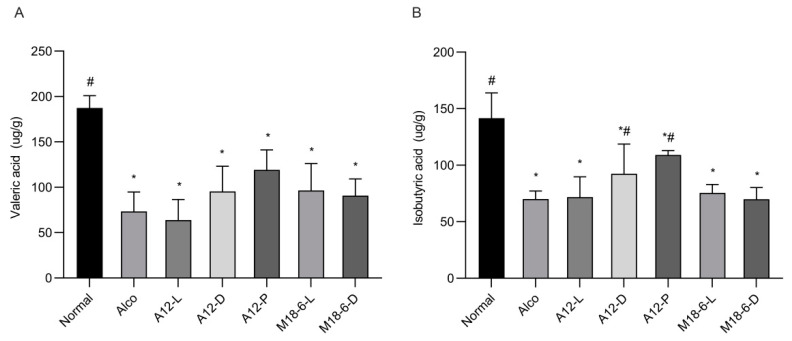
Short-chain fatty acid level in mice. (**A**): Valeric acid; (**B**): Isobutyric acid. Compared with normal group (*p* < 0.05) *, compared with Alco group (*p* < 0.05) #. Normal: normal saline-fed; Alco: alcohol-fed group; A12-L: alcohol-fed with live *B. animalis* A12 cells group, A12-D: alcohol-fed with dead *B. animalis* A12 cells group, A12-P: alcohol-fed with *B. animalis* A12 freeze-dried power group, M18-6-L: alcohol-fed with live *L. salivarius* M18-6 cells group, and M18-6-D: alcohol-fed with dead *L. salivarius* M18-6 cells group.

**Figure 7 foods-12-00439-f007:**
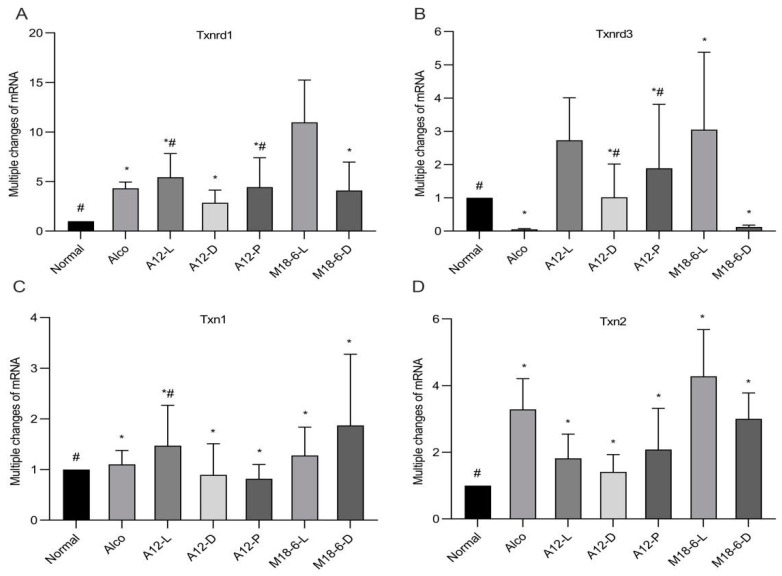
mRNA expression levels of thioredoxin system in livers of different groups. (**A**) Liver Txnrd1 mRNA expression level; (**B**) Liver Txnrd3 mRNA expression level; (**C**) Liver Txn1 mRNA expression level; (**D**) Liver Txn2 mRNA expression level. Compared with normal group (*p* < 0.05) *, compared with Alco group (*p* < 0.05) #. Txnrd1: thioredoxin reductase 1; Txnrd3: thioredoxin reductase 3; Txn1: thioredoxin 1; Txn2: thioredoxin 2. Normal: normal saline-fed; Alco: alcohol-fed group; A12-L: alcohol-fed with live *B. animalis* A12 cells group, A12-D: alcohol-fed with dead *B. animalis* A12 cells group, A12-P: alcohol-fed with *B. animalis* A12 freeze-dried power group, M18-6-L: alcohol-fed with live *L. salivarius* M18-6 cells group, and M18-6-D: alcohol-fed with dead *L. salivarius* M18-6 cells group.

**Figure 8 foods-12-00439-f008:**
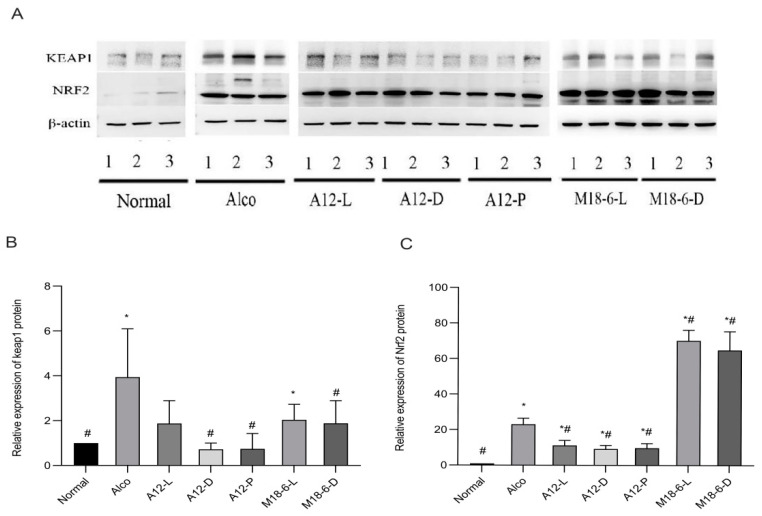
Expression of keap1/Nrf2 proteins in different treated groups. (**A**) Western blotting map. (**B**) Liver keap1 mRNA expression level; (**C**) Liver Nrf2 mRNA expression level. Compared with normal group (*p* < 0.05) #, compared with Alco group (*p* < 0.05) *. Keap1: Kelch-1ike ECH-associated protein l; Nrf2: Nuclear factor erythroid2-related factor 2. Normal: normal saline-fed; Alco: alcohol-fed group; A12-L: alcohol-fed with live *B. animalis* A12 cells group, A12-D: alcohol-fed with dead *B. animalis* A12 cells group, A12-P: alcohol-fed with *B. animalis* A12 freeze-dried power group, M18-6-L: alcohol-fed with live *L. salivarius* M18-6 cells group, and M18-6-D: alcohol-fed with dead *L. salivarius* M18-6 cells group.

**Table 1 foods-12-00439-t001:** Primers sequences used to amplify RNA transcripts.

Gene	Oligonucleotide (5′–3′)	Oligonucleotide (3′–5′)	Product(bp)
Keap1	TCGAAGGCATCCACCCTAAG	CTCGAACCACGCTGTCAATCT	135
Nrf2	TAGATGACCATGAGTCGCTTGC	GCCAAACTTGCTCCATGTCC	153
P65	ACTGCCGGGATGGCTACTAT	TCTGGATTCGCTGGCTAATGG	126
Txn1	GTGGTGTGGACCTTGCAAAA	CTGGCAGTCATCCACTCCA	100
Txnrd1	GGGTCCTATGACTTCGACCTG	AGTCGGTGTGACAAAATCCAAG	117
Txn2	TTCCCTCACCTCTAAGACCCT	CCTGGACGTTAAAGGTCGTCA	122
Txnrd3	GCACGCGGGTTAAGGAACT	TGGGCACCGTTTTCTGGTTAC	129

## Data Availability

The data are available from the corresponding author.

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
