# Peer review of "Bifidobacterium animalis A12 and Lactobacillus salivarius M18-6 Alleviate Alcohol Injury by keap1-Nrf2 Pathway and Thioredoxin System"

_foods, 2023, doi:10.3390/foods12030439_

Round 1

Reviewer 1 Report

Thank you for your manuscript.

I have no doubt about the quality of the presented work, but I recommend to revise the abstract so that the result appears more clearly. Also, please revise the Materials and Methods section. For example, the explanation for western blot can be shorter.  

A thorough English language review would improve the manuscript before acceptance for publication.

Line 39: add space after [1-4].

Line 48: remove starch

Line 68: B. animalis and L. salivarius should be in Italic.

Line 110 and 111: L. saliva should be L. salivarius

Line 134: change P65 to p65 (repeated in Line 258, 260, 263, 264, Figure 5)

Line 171: no need to abbreviate Eppendorf tube as EP.

Figure 2: Legend of the graph is barely visible. Please change the graph style.

Line 222: Normal should be in lowercase (same in Line 253, 333)  

Line 274-276. Need to re-sentence.

Also, at the end of the Discussion, please add limitations and future direction.

Author Response

We appreciate all the efforts you have to our manuscript. All the comments were very valuable to us. And had been carefully addressed in the revision. In the manuscript with tracked changes, the part marked with the red highlight was the revision following Reviewer 1’s advice. Moreover, we make some explaination to your questions.We hope our response to the comments satisfy reviewer.

Comments 1: Line 39: add space after [1-4].

Response 1: Thanks for your advice. We have added space after [1-4]. (Page 1 line 41).

Comments 2: Line 48: remove starch

Response 2: Thanks for your advice. We have removed starch. (Page 2 line 51-52).

Comments 3: Line 68: B. animalis and L. salivarius should be in Italic.

Response 3: We are very sorry for this oversight, we have changed B. animalis and L. salivarius to B. animalis and L. salivarius. (Page 2 line 71-72).

Comments 4: Line 110 and 111: L. saliva should be L. salivarius

Response 4: As suggested by the Reviewer, we have changed the word “L. saliva” to “L. salivarius”. (Page 3 line 117).

Comments 5: Line 134: change P65 to p65 (repeated in Line 258, 260, 263, 264, Figure 5)

Response 5: As it is suggested, we have changed the word “P65” to “p65”. (Page 4 line 145; Page 8 line 287, 289, 292, 294).

Comments 6: Line 171: no need to abbreviate Eppendorf tube as EP.

Response 6: As it is suggested, we have changed the word “EP” to “Eppendorf”. (Page 5 line 182).

Comments 7: Figure 2: Legend of the graph is barely visible. Please change the graph style.

Response 7: we have changed Figure 2 style. (Page 6 line 232).

Comments 8: Line 222: Normal should be in lowercase (same in Line 253, 333)  

Response 8: line : we have changed the word “Normal” to “normal”. (Page 7 line 246; Page 8 line 278).

Comments 9: Line 274-276. Need to re-sentence. 

Response 9: Thanks very much for your valuable comments, we have re-sentenced as short-chain fatty acids (SCFA) could affect intestinal permeability and maintain normal physiological function of the gut. (Page 9 line 309-310).

Comments 10: Also, at the end of the Discussion, please add limitations and future direction.

Response 10: We think the comment are very desirable, also, we have added limitations and future direction. (Page 13 line 482-486).

Reviewer 2 Report

The study analyzed the effect of Bifidobacterium animalis A12 and Lactobacillus salivarius M18-6 on alcohol injury by assessing the keap1-Nrf2 and thioredoxin pathways. The study was well-carried and current form; I have some issues that need to be addressed.

The main limitation of the study is the lack of gut microbiota 16S seq. The authors should include the main limitation in a revised version.

Introduction

Oxidative Stress Tolerance and Anti‑Oxidant Capacity in probiotic strains should be better displayed in the introduction (DOI: 10.1007/s12602-022-09943-3).

Lines 78-79: please add a reference to this information.

Methods

What number of animals were used in each experiment? Please, add this information in a revised version (text and tables).

Please, in each figure footnote, add the meaning of abbreviations.

The authors used SPPS for statistical analyses, but another program made the graphics. Please, if I’m correct, clarify it in the manuscript.

The authors used symbols and letters to identify the statistical difference in the figures. The authors should standardize it.

Results

Please, add the exact p-value in the description of the results.

Please, remove the discussion and reference citation in the results section.

Lines 224-226: What (Figure 3A. b), (Figure 3A. f, g), (Figure 3A. d)?

Lines 263-264: what (1.26) and (0.99)?

Minor

Native English could revise the manuscript.

Author Response

Thank you so much for your detailed and comprehesive advice. In the revision manuscript with tracked changes, the part marked with the yellow highlight was the revision following Reviewer 2’s advice. Moreover, we make some explaination to your questions.We hope our response to the comments satisfy reviewer.

The number of page and line shown in “response” is marked in the manuscript with Tracked Changes.

Comments 1: The main limitation of the study is the lack of gut microbiota 16S seq. The authors should include the main limitation in a revised version.

Response 1: Thanks for your advice. We are very sorry for the lack of gut microbiota 16S seq in the study. we will further study the gut microbiota and more in-depth discussion in the future.

Comments 2: Lines 78-79: please add a reference to this information.

Response 2: We are very sorry for this oversight, we have added a reference to this information. Dong, C.; Zhang, H.; Jia Y.; Xie, Y.; Liu, H.; Jin, J. Evaluation of antioxidative function of Lactobacillus salivarius M18-6 in vitro and its antioxidant mechanisms[J]. Food and Fermentation Industries, 2021, 47 ,132-137. (Page 2 line 82-83).

Comments 3: What number of animals were used in each experiment? Please, add this information in a revised version (text and tables).

Response 3: Thanks for your advice. We have added animals number in text and tables. (Page 3 line 113; Page 5 line 212-213).

Comments 4: Please, in each figure footnote, add the meaning of abbreviations.

Response 4: According to your comment, we have added the meaning of abbreviations in each figure footnote. (Page 6 line 235-239; Page 7 line 262-266; Page 8 line 279-283; Page 9 line 303-307; Page 9 line 319-323; Page 10 line 346-351; Page 11 line 373-378, Page 14 line 514-522).

Comments 5: The authors used SPPS for statistical analyses, but another program made the graphics. Please, if I’m correct, clarify it in the manuscript.

Response 5: Yes, you are right, we used SPSS for statistics analyses and GraphPad Prism 8 for graphics, we have clarified in the manuscript. (Page 5 line 210-211).

Comments 6: The authors used symbols and letters to identify the statistical difference in the figures. The authors should standardize it.

Response 6: We think the comment are very desirable. We have standardized the used symbols and letters to identify the statistical difference of all figures. Compared with normal group (P < 0.05)*, compared with Alco group (P < 0.05) #.

Comments 7: Please, add the exact p-value in the description of the results.

Response 7: According to your comment, we have added the exact p-value in the description of the results. (Page 7 line 255, 270, 272; Page 8 line 290, 293, 297; Page 9 line 311, 314, 316, 329, 330, 337).

Comments 8: Please, remove the discussion and reference citation in the results section. 

Response 8: As it is suggested, we have removed the discussion and reference citation in the results section. 

Comments 9: Lines 224-226: What (Figure 3A. b), (Figure 3A. f, g), (Figure 3A. d)?

Response 9: We have re-written this part according to the Reviewer’s comment. We have renumbered Figure 3 to make it more concise and easy to understand. (Page 7 line 248,250-251).

Comments 10: Lines 263-264: what (1.26) and (0.99)?

Response 10: (1.26) shows the p65 mRNA change folds in the A12-L group. (0.99) shows the p65 mRNA change folds in the M18-6-D group. (Page 8 line 293).

Comments 11: Native English could revise the manuscript.

Response 11: Our manuscript has been revised by native English. (Page 13 line 501).

Reviewer 3 Report

In the present study, the authors have explored the preventing effect against alcohol-induced hepatic injury in mice of probiotic strains. They have also analyzed some related indicators and formed a hypothesis based on their results.

The study seems to be interesting and well designed and conducted.  However, I have some questions and proposals about the result as follows to make the manuscript superior.

Major

1. Figure 2 can be improved to indicate the difference. Further, the authors should analyze the statistical difference among the groups by a log-rank test. If they are not sure, please refer to the following reference.

Int. J. Mol. Sci. 2020, 21, 1896; doi:10.3390/ijms21051896

2. It seems difficult to lead the conclusion that probiotics used in the present study were effective in preventing alcohol injury from the results provided by Figure 3B and Figure 4. Please provide more detailed explanation and/or discussion.

3. Figure 5 (SOD) seems to indicate that powder form of strain A12 are superior to other forms of it (also written at line 366), and Figure 6 are too. Do the authors have any thoughts and ideas to explain the different effectiveness between type of forms?

Minor

1. The authors should provide the approval no. of the ethical committee.

2. The centrifugal conditions should be written using ×g instead of rpm (with temperature).

3. Primer sequences should be presented as manner, 5’- NNNN -3’.

4. Soft version no. of SPSS should be provided.

5. The genus and species are always written in italics.

Author Response

Thank you so much for your detailed and comprehesive advice. In the revision manuscript with tracked changes, the part marked with the blue highlight was the revision following Reviewer 3’s advice. Moreover, we make some explaintion to some questions. We hope our response to the comments satisfy reviewer.

The number of page and line shown in “response” is marked in the manuscript with Tracked Changes.

Major comments

Comments 1: Figure 2 can be improved to indicate the difference. Further, the authors should analyze the statistical difference among the groups by a log-rank test. If they are not sure, please refer to the following reference.

Int. J. Mol. Sci. 2020, 21, 1896; doi:10.3390/ijms21051896

Response 1: Thanks for your advice. We have improved Figure2 and analyzed the statistical difference among the groups by a log-rank test. (Page 6 line 232-233)

Comments 2: It seems difficult to lead the conclusion that probiotics used in the present study were effective in preventing alcohol injury from the results provided by Figure 3B and Figure 4. Please provide more detailed explanation and/or discussion.

Response 2: As it is suggested, we have provided more detailed explanation and discussion for Figure 3B and Figure 4. As shown in Figure 3B, liver index in the Alco group was lower compared to the normal group, the differences were not significant (P>0.05). Therefore, we examined multiple indicators to prove that probiotics used in the present study can effectively alleviate alcohol injury. Serum ALT was considered as biochemical markers of liver injury. As shown in Figure 4, compared with the normal group, the serum ALT in the Alco group was significantly increased, B. animalis A12-L and L. salivarius M18-6-D significantly reversed serum ALT due to alcohol injury (P<0.05). (Page 7 line 251-257; Page 7 line 269-276)

Comments 3: Figure 5 (SOD) seems to indicate that powder form of strain A12 are superior to other forms of it (also written at line 366), and Figure 6 are too. Do the authors have any thoughts and ideas to explain the different effectiveness between type of forms?

Response 3: Based on the reviewer's comments, we further analyzed and compared the differential effectiveness between the different forms of the strain. (Page 11 line 381-388)

Live cells are beneficial to the host, it can survive and colonize in the animal digestive tract, compete with pathogenic bacteria, and produce useful enzymes and metabolites, after inactivation, the active components of the cell wall, such as exopolysaccharides, peptidoglycan and lipoteichoic acid, will be dissociated from the cell wall membrane, and may play an active role in some functions. Freeze-dried powder uses multi-layer embedding technology to have high survival rate, protect strains from reaching the intestine alive, adhere to the intestine, and drive away harmful bacteria. Since different forms may have different effects, they are worth discussing.

Minor comments

Comments 1: The authors should provide the approval no. of the ethical committee.

Response 1: As suggested by the Reviewer, we have supplemenented the approval no. of the ethical committee. (Page 3 line 106-107; Page 14 line 511)

Comments 2: The centrifugal conditions should be written using ×g instead of rpm (with temperature).

Response 2: According to the suggestion, we have changed “rpm” to “×g”. (Page 4 line 140)

Comments 3: Primer sequences should be presented as manner, 5’- NNNN -3’.

Response 3: We have re-written this part according to the Reviewer’s comment. (Page 4 line 158)

Comments 4: Soft version no. of SPSS should be provided.

Response 4: As suggested by the Reviewer, we have provided Soft version of SPSS, it should be IBM SPSS Statistics 26. (Page 5 line 206)

Comments 5: The genus and species are always written in italics.

Response 5: We are very sorry for this oversight, we have corrected the text.

Round 2

Reviewer 2 Report

Theo authors addressed all my early concerns. 

Reviewer 3 Report

The revised manuscript seems to be improved.

Minor:

The both sense- and anti-sense primers used in the study should be written as not 3'-NNNNNN-5', but 5'-NNNNNN-3' manner.